# PT320, a Sustained-Release GLP-1 Receptor Agonist, Ameliorates L-DOPA-Induced Dyskinesia in a Mouse Model of Parkinson’s Disease

**DOI:** 10.3390/ijms24054687

**Published:** 2023-02-28

**Authors:** Tung-Tai Kuo, Yuan-Hao Chen, Vicki Wang, Eagle Yi-Kung Huang, Kuo-Hsing Ma, Nigel H. Greig, Jin Jung, Ho-II Choi, Lars Olson, Barry J. Hoffer, Kuan-Yin Tseng

**Affiliations:** 1National Defense Medical Center, Taipei 11490, Taiwan; 2Department of Neurological Surgery, Tri-Service General Hospital, Taipei 11490, Taiwan; 3Ph.D. Program in Translational Medicine, National Defense Medical Center and Academia Sinica, Taipei 11490, Taiwan; 4Department of Pharmacology, National Defense Medical Center, Taipei 11490, Taiwan; 5Graduate Institute of Biology and Anatomy, National Defense Medical Center, Taipei, 11490, Taiwan; 6Drug Design & Development Section, Translational Gerontology Branch, Intramural Research Program National Institute on Aging, National Institutes of Health (NIH), Baltimore, MD 21224, USA; 7Peptron, Inc., Yuseong-gu, Daejeon 34054, Republic of Korea; 8Department of Neuroscience, Karolinska Institute, 171 77 Stockholm, Sweden; 9Department of Neurosurgery, University Hospitals of Cleveland, Case Western Reserve University School of Medicine, Cleveland, OH 44106, USA

**Keywords:** PT320, L-DOPA-induced dyskinesia, Parkinson’s disease, MitoPark, dopamine

## Abstract

To determine the efficacy of PT320 on L-DOPA-induced dyskinetic behaviors, and neurochemistry in a progressive Parkinson’s disease (PD) MitoPark mouse model. To investigate the effects of PT320 on the manifestation of dyskinesia in L-DOPA-primed mice, a clinically translatable biweekly PT320 dose was administered starting at either 5 or 17-weeks-old mice. The early treatment group was given L-DOPA starting at 20 weeks of age and longitudinally evaluated up to 22 weeks. The late treatment group was given L-DOPA starting at 28 weeks of age and longitudinally observed up to 29 weeks. To explore dopaminergic transmission, fast scan cyclic voltammetry (FSCV) was utilized to measure presynaptic dopamine (DA) dynamics in striatal slices following drug treatments. Early administration of PT320 significantly mitigated the severity L-DOPA-induced abnormal involuntary movements; PT320 particularly improved excessive numbers of standing as well as abnormal paw movements, while it did not affect L-DOPA-induced locomotor hyperactivity. In contrast, late administration of PT320 did not attenuate any L-DOPA-induced dyskinesia measurements. Moreover, early treatment with PT320 was shown to not only increase tonic and phasic release of DA in striatal slices in L-DOPA-naïve MitoPark mice, but also in L-DOPA-primed animals. Early treatment with PT320 ameliorated L-DOPA-induced dyskinesia in MitoPark mice, which may be related to the progressive level of DA denervation in PD.

## 1. Introduction

L-DOPA (L-3,4-dihydroxyphenylalanine) is considered to be the standard treatment for Parkinson’s disease (PD); however, chronic administration often causes abnormal involuntary movements (AIMs), also called L-DOPA-induced dyskinesia (LID), in PD patients [1,2]. Although the mechanisms underlying the generation of LID are complex, some studies suggest that the development and severity of dyskinesia correlates with the degree of dopamine (DA) loss and the dose of L-DOPA administered rather than the duration of L-DOPA therapy [3,4,5,6,7]. Furthermore, aberrant release and clearance of exogenous L-DOPA in non-dopaminergic terminals within the dopamine-denervated striatum might contribute to anomalous regulation of motor behavior [8]. Currently, several compounds that regulate various neurotransmitter systems (such as the 5-HT, ACh or GABA systems, which all interact with dopaminergic pathways) have been shown to reduce the severity of LID [9,10]. Preclinical studies suggest that anti-inflammatory compounds such as corticosterone, cannabidiol and thalidomide may also be beneficial in attenuating LID by inhibiting the inflammatory activation of microglia [11]. However, the outlook for such interventions remains limited, with different regulatory agencies claiming that there is insufficient evidence to support implementing these preliminary data in clinical practice. In this study, the MitoPark mouse, with a homozygous disruption of mitochondria transcription factor A (TFAM) specifically in midbrain DA, was adopted to characterize PD’s neurofunctional impairment [12]. Given its slow, progressive loss of DA, the MitoPark mouse has been studied as a preclinical animal model to recapitulate LID in response to DA replacement therapy [13,14]. 

Recent studies have emerged supporting the notion that glucagon-like peptide-1 (GLP-1), as well as long-acting GLP-1 receptor (GLP-1R) agonists approved for the treatment of type 2 diabetic mellitus (T2DM), may also be used as new non-DA replacement anti-Parkinsonian drugs by mitigating DA neuron degeneration [15,16]. However, the clinical utility of GLP-1R agonists is limited by their relatively short half-life and poor brain uptake [15,17]. Our prior studies have demonstrated that the application of exendin-4, a non-hydrolysable agonist for GLP-1, provides good brain uptake in humans; a therapeutic level is achieved by steady-state, long-term administration of the drug [15,18,19]. In addition, the application of sustained-release technology for exendin-4 provides a sustained-release formulation termed PT320, which maintains steady-state plasma levels and brain concentrations to optimize drug treatment for the slow progression of debilitating neurodegenerative disorders [15,19,20,21]. 

We and others have previously demonstrated that subcutaneous administration of PT320 significantly counteracts behavioral deficits in various animal models of PD [15,16,20,22]. The beneficial action of PT320 treatment is attributed to mitigation of dopaminergic neurodegeneration. In addition, PT320 treatment was shown to slow loss of striatal and nucleus accumbens DA release and reuptake in the MitoPark model [22], suggesting that PT320 treatment promotes neuroprotective activity in DA pathways. 

The purpose of this study was to determine the effects of PT320 on abnormal involuntary movements in L-DOPA-responsive, DA-deficient MitoPark mice. We designed a therapeutic regimen in which PT320 was administered starting at 5 weeks of age, in accordance with a previously defined time course, and in which L-DOPA/carbidopa treatment started at 20 weeks of age and continued for two weeks. We found that this dosing schedule attenuated LID development. Notably, neurochemical analysis revealed that PT320 also preserved the capacity of evoked DA release in the DA-denervated striatum. This pharmacological approach to suppress dyskinesia in MitoPark mice is correlated with preservation of quantal DA release by PT320 treatment to reduce LID development in L-DOPA-primed animals. In contrast, late initiation of PT320 treatment, starting at 17 weeks, did not reduce LID in MitoPark mice evaluated at 28 weeks. 

## 2. Results

### 2.1. L-DOPA Priming

The severity of LID was assessed according to the AIM score system, which includes degree of twitching of the trunk, latency of body elevation, severity of limb displacement an number of abnormal movements (Appendix A). We first investigated whether older MitoPark mice without L-DOPA replacement gradually develop dyskinetic behavior assessed by the AIM score system. As shown in Appendix A, AIM scores did not increase in 22-week-old and 29-week-old MitoPark mice without L-DOPA treatment compared to age-matched WT mice, respectively. Furthermore, administration of PT320 starting at 5 weeks of age (early treatment) or at 17 weeks of age (late treatment) did not affect AIM scores analyzed in 22-week-old or 29-week-old MitoPark mice, suggesting that MitoPark mice do not develop dyskinetic behavior either when they become older or only receive PT320 treatment. Next, we investigated whether the development of LID depended on the stage of the PD at initiation of L-DOPA treatment. 28-week-old MitoPark mice exhibited more severe LID, as well as a longer response duration of the L-DOPA injection, when compared to 20-week-old MitoPark mice (Figure 1A–C). Thus, there were lower overall AIM scores in 20-week-old MitoPark mice on days 1, 4 and 7 after L-DOPA/carbidopa treatment (Figure 1D). These data suggest that both the extent of DA degeneration and subsequent adaptions in the striatum determine the severity of the dyskinesia that develops after initiation of L-DOPA replacement therapy.

### 2.2. Early Administration of PT320 Reduced L-DOPA-Induced Dyskinesia 

Next, we examined if early administration of PT320 could reduce dyskinetic development in MitoPark mice after 2 weeks of treatment with L-DOPA/carbidopa. Based on a previously established protocol, PT320 was administered starting at 5 weeks of age (early treatment with PT320); the timeline of biweekly PT320 treatments is shown in Figure 2A. Our prior studies involving the pharmacokinetics of PT320 in mice demonstrated that the selected dose of 30 mg/kg is of translational clinical relevance [21,22]. The LID induction protocol (L-DOPA/carbidopa twice a day for 2 weeks) was initiated in MitoPark mice at 20 weeks of age. Over the 14-day LID induction, increased overall AIM scores were found on day 4 (Figure 2B), suggesting that 20-week-old MitoPark mice develop early dyskinesia under L-DOPA/carbidopa treatment. Although early treatment with PT320 might lower the occurrence of LID on day 1 after initiation of L-DOA/carbidopa treatment (Appendix A), it did not initially suppress overall dyskinesia development during the early period of LID induction (Appendix A). Importantly, however, PT320 did significantly reduce overall AIM scores in L-DOPA-treated MitoPark mice after 2-weeks (Figure 2C and Appendix A), and also decreased the AIM score measured at 100 min after L-DOPA injection (Appendix A), suggesting that this strategy may accelerate the recovery process from abnormal involuntary movements following L-DOPA injection. In addition, the slope of AIM scores during the LID induction period was less in the early PT320 treatment group (Figure 2D) than in the vehicle group (L-DOPA-primed MitoPark; MP LID). Taken together, this PT320 treatment protocol could gradually alleviate the severity of dyskinesia during the period of LID induction and consequently attenuate LID in 2-week L-DOPA-primed MitoPark mice. 

To further clarify which L-DOPA-induced abnormal behaviors were primarily affected by early treatment with PT320, differences in each component of the AIM score system between the PT320 treatment and control groups of 2-week L-DOPA-primed MitoPark mice were analyzed. Abnormal axial deviation was alleviated at 100 min after a single L-DOPA injection in the PT320 treatment group (Figure 3A); however, there were no differences in overall axial or limb AIM scores in the dyskinetic period of 20 to 120 min after a L-DOPA injection between the two groups (Figure 3B–D). Notably, however, PT320 treatment significantly reduced abnormal standing and suppressed abnormal paw movements at late recording time points (Figure 3E–H). Overall standing or paw movement AIM scores were significantly decreased in the PT320 treatment group during the observation period (Figure 3F,H). Taken together, this PT320 therapeutic protocol was shown to predominantly suppress both abnormal standing behavior and abnormal paw movements in L-DOPA-primed MitoPark mice.

### 2.3. Early Administration of PT320 Did Not Alleviate L-DOPA-Induced Locomotor Hyperactivity

Next, we investigated if early PT320 treatment could also modify locomotor activity in L-DOPA-primed MitoPark mice. Locomotor activities, including ambulation distance and rearing duration, were analyzed for 60 min following L-DOPA injection (Figure 4A). Consistent with our previous studies, 22-week-old MitoPark mice exhibited lower locomotor activity compared to age-matched WT mice. However, at the same time point PT320-treated animals did not show better horizontal locomotion or rearing activity than untreated MitoPark mice (Figure 4B,C). Following the 14-day administration of L-DOPA/carbidopa, both the horizontal travel distance and rearing duration of LID MitoPark mice (MP LID) were longer than those of WT or MitoPark mice (Figure 4B,C), implying that L-DOPA-primed MitoPark mice develop abnormal locomotor behavior. However, early treatment with PT320 did not alleviate L-DOPA-induced locomotor hyperactivity (Figure 4B,C). These data suggest that the increased sensitivity to L-DOPA administration shown in L-DOPA-primed MitoPark mice could not be alleviated by our PT320 treatment regimen. 

### 2.4. Early Administration of PT320 Increases Tonic and Phasic DA Release in Striatum of MitoPark LID Mice after L-DOPA Administration 

In line with our previous study, administration of levodopa increased DA release in the dorsal striatum of the MitoPark LID group (Figure 5A,B). Notably, the release of DA following L-DOPA infusion was larger in the MitoPark mice receiving early treatment of PT320 than in untreated mice (Figure 5C,D). During tonic or phasic stimulation, the release of DA in PT320-treated L-DOPA-primed MitoPark mice actually increased compared to that of the MitoPark LID group (Figure 5A,B). The average concentration of tonic or phasic (bursting) DA release after L-DOPA administration was higher in the L-DOPA-primed MitoPark mice receiving early PT320 treatment than in the MitoPark LID group (Figure 5C,D). Taken together, early treatment with PT320 did not attenuate increments of tonic and phasic DA release induced by L-DOPA, which is related to LID development. Instead, this regimen of PT320 treatment might preserve the tonic or phasic DA release in the DA-denervated striatum exposed to L-DOPA. 

### 2.5. Late Administration of PT320 Did Not Mitigate L-DOPA-Induced Dyskinesia 

Next, we investigated whether PT320 treatment could also ameliorate dyskinesia development in older MitoPark mice with severe DA degeneration. Briefly, PT320 was administered starting at 17 weeks of age (late treatment of PT320). The timeline of biweekly PT320 treatments is shown in Figure 6A. In this regimen of PT320 treatment, MitoPark mice exhibited decreased body weight beginning at an age of 20 weeks, but there was no difference in the amount of either food intake or locomotor activity between the PT320-treated and untreated groups. The LID induction protocol (L-DOPA/carbidopa twice a day for 1 week) was performed in MitoPark mice starting at 28 weeks of age. In the 7-day period of LID induction, increased overall AIM scores were found on day 1 (Figure 6B), suggesting that the development of LIDs in 28-week-old MitoPark mice was abrupt and marked even after the very first L-DOPA injection. The late administration of PT320 neither suppressed dyskinesia development during the period of LID induction (Appendix A) nor reduced overall AIM scores on day 7 after initiation of L-DOPA/carbidopa treatment (Figure 6B,C; Appendix A). There was no significant difference in the slope of AIM scores in the LID induction period between the late PT320-treated and untreated groups (Figure 6D). To better determine if late PT320 treatment merely affects one subtype of L-DOPA-induced abnormal behaviors, we analyzed each item of the AIM scores in the dyskinetic period of 20 to 160 min after L-DOPA injection. 

There was no difference in severity of axial deviation (Figure 7A,B), extent of limb displacement (Figure 7C,D), intensity of abnormal paw movement (Figure 7G,H) or duration of standing (Figure 7E,F) between late PT320-treated and untreated groups. 

In addition, late PT320 treatment did not promptly suppress LID development in 29-week-old MitoPark mice after the first L-DOPA injection, in contrast to early PT320 treatment in 22-week-old mice (Figure 8A). Furthermore, late PT320 treatment did not shorten the duration of dyskinetic movements following L-DOPA injection on days 4 and 7 after initiation of L-DOPA/carbidopa treatment, as was shown with early PT320 treatment (Figure 8B,C). Similarly, late PT320 treatment did not manifest any therapeutic effects on LID manifestation in 29-week-old MitoPark mice with advanced DA degeneration (Figure 8D).

## 3. Discussion

We found that early administration of PT320 significantly mitigated the severity of L-DOPA-induced abnormal involuntary movements. In particular, this early administration improved excessive amounts of standing as well as abnormal paw movements, whereas it did not affect L-DOPA-induced locomotor hyperactivity. In contrast, late administration of PT320 did not attenuate L-DOPA-induced dyskinesia measurements. Interestingly, early treatment with PT320 was shown to not only increase tonic and phasic release of DA in striatal slices from MitoPark mice [22] but also in L-DOPA-treated animals.

We did not include additional groups of untreated MP mice. As noted below in Methods, the MP mice used here were divided from only heterozygote matings. This is because homozygous MP mice are subfertile. As a result, only one-fourth of a litter are MP mice as confirmed by genotyping. Thus, in order to obtain enough animals for the treatment groups used here, untreated groups were not included. In our experience [12,13,23,24,25], aging alone manifests severe bradykinesia and weight loss in MP mice but not dyskinesia, a confound at more than 30 weeks of age.

Thus, early PT320 treatment could ameliorate L-DOPA-induced dyskinesia in MitoPark mice, which may be related to the progression level of DA denervation in PD. Disruption of mitochondrial transcription factor A (TFAM) in dopamine neurons of MP mice causes progressive impairment and swelling of mitochondria, as well as intraneuronal inclusions. Both the number and size of these inclusions in cell bodies and nerve processes of dopamine neurons were observed from 6 to 30 weeks of age in MP mice. The internal structure of the mitochondria (cristae) is progressively lost as well [23].

Several prior studies, including ours, have demonstrated that GLP-1R agonists, exendin-4 (i.e., exenatide) in particular, augment not only the survival of primary neuronal cell cultures but also that of immortal neuronal cell lines challenged with a variety of toxic insults, including 6-hydoxydopamine (6-OHDA), oxidative stress, α-synuclein, amyloid-β peptide, glutamate excitotoxicity, and hypoxia [15,16,26,27,28,29,30]. These neuroprotective actions appear to translate to rodent preclinical animal models of neurogenerative disorders; GLP-1R agonists have demonstrated efficacy in rodent models of PD, Alzheimer’s disease, ischemic stroke, traumatic brain injury, peripheral neuropathy, amyotrophic lateral sclerosis, Huntington’s disease and idiopathic intercranial hypertension [15,16,31,32,33]. Likewise, neurotrophic GLP-1R-mediated actions have been demonstrated across both neuronal in vitro and in vivo models, including the development of a stronger cellular phenotype, as evaluated by an increased level of tyrosine hydroxylase immunoreactivity in unchallenged primary dopaminergic cell cultures [28], of choline acetyl transferase in cholinergic cells cultures [30], and the induction of neurite outgrowth [27]. Trophic support is essential for neuronal cell health and survival [34], and these neuroprotective actions of GLP-1R agonists appear to be mediated via the protein kinase cAMP-dependent (*PKA*) and phosphoinositide 3-kinase (*PI3K*)-AKT pathways, as their selective inhibition results in a loss of GLP-1- mediated actions [15,27,30]. Anti-inflammatory actions have additionally been demonstrated for GLP-1R agonists across cellular and animal models of neurodegeneration [15,16,32,33,35] and largely involve the same pathways. Recent studies in patients with moderate PD have demonstrated that beneficial GLP-1R-mediated actions translate to humans [18] with actions being mediated via brain insulin-signaling proteins, Akt and mechanistic target of rapamycin (mTOR) pathways. Favorable actions of GLP-1R agonists have similarly been reported to occur via improvements in mitochondrial function [16,36]. It is quite likely that a combination of effects augmenting neurotrophic, neuroprotective, anti-inflammatory, mitochondrial and potentially other actions is responsible for the beneficial activity of early PT320 treatment in mitigating LID in MitoPark mice in our study.

Our data on age-dependent LID development in animals confirm and extend previous studies in MitoPark mice [12,13,24,25]; there is also clinical literature suggesting this as well. The rapid emergence of dyskinesia in PD patients with either a late diagnosis or early onset, where denervation is extensive at diagnosis, as well as the absence of dyskinesia in normal humans chronically treated with L-DOPA (i.e., mistaken diagnosis) [37,38,39] support this conjecture.

Precisely why early administration of a long-acting, slow-release GLP-1R agonist like PT320 slows the development of LIDs requires further studies. However, our recent paper showing that PT320 augments nigrostriatal function behaviorally and physiologically, as well as slowing loss of DA markers in MitoPark mice, suggest that slowing the loss of the nigrostriatal DA input is a critical PT320 mechanism [22]. Interestingly, a study of PT320 in the 6-OHDA rat model of PD also showed a mitigation of LIDs by PT320 [40]. The rapid development of nigrostriatal DA denervation by 6-OHDA precluded the temporal early vs late PT320 exposure protocol in that model, in contrast to the results reported here.

## 4. Materials and Methods

### 4.1. Animals

MitoPark mice used in this study were homozygous for the loxP-flanked Tfam gene (Tfam^loxP^/Tfam^loxP^) and heterozygous for DAT-cre (DAT/DAT^cre^). Breeding pairs to generate MitoPark mice and genotyping were performed at the National Laboratory Animal Center of the National Applied Research Laboratories in Taiwan. The breeding scheme for generating MitoPark mice has been described in previous studies [12,13,23]. 5-week-old and 17-week-old male MitoPark mice were housed at 25 °C with a 12/12 light/dark cycle, as well as a sufficient food and water supply, in the National Defense Medical Center (NDMC) animal facility, which has full AAALAC accreditation. All experimental animal protocols were approved by the NDMC Animal Care and Use Committee (IACUC 20-129 and IACUC 22-008) following National Institutes of Health guidelines. See Figure 2A, Figure 6A, and Figure 7A for experimental protocols and times of drug administration. MitoPark mice have difficulty eating when they are older than 20 weeks. Therefore, the nutritionally fortified dietary supplement DietGel^®^ 76A, contained in MFG feeding and drinking water, was used starting from 20 weeks of age.

### 4.2. Treatments

The MitoPark mice were randomly divided into groups with or without PT320 treatment at 5 or 17 weeks. Experimental animals were administered 30 mg/kg PT320 [20,21] subcutaneously every two weeks, starting at 5 or 17 weeks of age. Our prior pharmacokinetic studies indicate that this 30 mg/kg PT320 dose is of clinical relevance [22]. PT320 comprises 2% Exenatide and 98% polymers from which Exenatide is slowly released. The total amount of material dispensed every two weeks was 30 mg/kg PT320, which is equivalent to 0.6 mg/kg Exenatide. As the PT320 microsphere powder does not dissipate into a solution but rather generates a suspension, the preparation was ‘vortexed’ immediately prior to each injection, with verification that the PT320 material was in suspension prior to administration.

10 mg/kg L-DOPA (Sigma-Aldrich, St. Louis, MO, USA) and 5 mg/kg carbidopa (Sigma-Aldrich, St. Louis, MO, USA) were dissolved in 0.9% normal saline. L-DOPA was injected intraperitoneally together with carbidopa to inhibit peripheral dopa decarboxylase. L-DOPA and carbidopa were injected twice daily (at 08:00 and 16:00). The first L-DOPA administration protocol was started in the MitoPark mice at 20 weeks and continued for 14 days to induce dyskinesia, as shown in Figure 2A. The second L-DOPA administration protocol was started in the MitoPark mice at 28 weeks and continued for seven days, as shown in Figure 7A. The protocol for using administration of L-DOPA to induce an age-dependent MP mouse generation of LID is also detailed in our previous papers [12,13,24,25].

### 4.3. Behavioral Tests

**Abnormal involuntary movements (AIM)**. AIM scoring was used to measure the severity of dyskinesia in MitoPark mice after the injection of L-DOPA and carbidopa. The AIM scoring used was a modified version of the mouse AIM behavior grading proposed previously [41,42]. The modified AIM scoring standards are shown in Appendix A. In the first protocol, MitoPark mice were observed for 120 min after administering 10 mg/kg L-DOPA and 5 mg/kg carbidopa on days 1, 4, 7, and 14. In the second protocol, MitoPark mice were observed for 160 min after the administration of 10 mg/kg L-DOPA and 5 mg/kg carbidopa on days 0, 1, 2, 3, 4, and 7. A webcam was used to simultaneously record each mouse’s behavior for 1 min every 20 min. Dyskinesia severity was recorded every 20 min by blinded observers, with the AIM score defined as the most severe symptom according to the above-mentioned scoring standard. The higher the AIM score, the more abnormal involuntary movement behaviors were noted. For untreated mice, the total AIM score should be below 4 [43].

**Locomotor activity.** Activity boxes (45 × 45 cm) were used to evaluate horizontal and vertical movements of MitoPark mice after habituation in a low-noise environment for 1 h. The protocol used a grid of infrared beams to monitor both the horizontal movement distance and the number of rearings over the 1-h period.

### 4.4. Fast Scan Cyclic Voltammetry (FSCV) for DA Dynamic Measurements in Brain Slices

Striatal Brain Slice Preparation. Details of the brain slice preparation process have been described in our previous studies [44,45], and are briefly presented here. After mice were sacrificed by decapitation, their brains were removed and transferred to the cutting solution (in mM: sucrose 194, NaCl 30, KCl 4.5, MgCl_2_ 1, NaH_2_PO_4_ 1.2, glucose 10, and NaHCO_3_ 26) with oxygenation (95%O_2_/5%CO_2_). After 30 s, the brain tissue was separated by a vibrating blade microtome into coronal 280 μm slices containing dorsal striatum (VT 100, Leica) in a chamber filled with the cutting solution and oxygenation (95%O_2_/5%CO_2_). Finally, the brain slices were transferred to a chamber with a solution containing oxygenated artificial cerebrospinal fluid (aCSF; in mM: NaCl 126, KCl 3, MgCl_2_ 1.5, CaCl_2_ 2.4, NaH_2_PO_4_ 1.2, glucose 11, NaHCO_3_ 26) at 30 °C for 30 min and subsequently used in the FSCV experiments.

**Fast scan cyclic voltammetry (FSCV) for DA measurements in brain slices.** Detailed FSCV protocols used in this study have been described in our previous papers [46,47]. The brain slices from the 22-week-old mice were moved into the chamber (0.5 mL, 31–33 °C) filled with aCSF and with a perfusion rate of 2 mL/min for DA measurements. A carbon fiber electrode consisting of a carbon fiber (7 μm diameter; Goodfellow Corp., Oakdale, PA, USA) inside a pipette filled with 150 mM KCl solution and 4M potassium acetate was used for FSCV measurements. The carbon fiber electrode was positioned between the separated tips of the bipolar stimulation electrodes (FHC Inc., Bowdoin, ME, USA) in the dorsal striatum at a depth of 100 μm for DA measurements. The FSCV used a triangular waveform (400 V/s scan rate, 7 ms duration) applied every 100 ms to drive the potential of the carbon fiber from −0.4 to 1.0 V and back to −0.4 V, using a 5 V stimulus intensity at 25 Hz to induce DA release. Data collection and post-mortem analysis were carried out using an A/D board (PCI 6052E and PCI-6711E, National Instruments, Austin, TX, USA) and customized software based on LabView (TarHeel CV, courtesy of Drs. Joseph Cheer and Michael Heien, University of North Carolina).

DA release in striatal slices was detected by fast scan cyclic voltammetry (FSCV) in wild type (WT), MitoPark, MitoPark LID, and MitoPark LID + PT320 mice. To examine the effect of L-DOPA on the ability to release DA in the striatum, the DA precursor L-DOPA (0.1 μM) was administered for 30 min into striatal slices followed by a washout period of 120 min. DA release was induced by 5 V electrical stimulation every two minutes to ensure DA neurons recovered from the previous electrical stimulation described above. Each carbon fiber electrode was calibrated with a 0.1 μM DA standard solution before and after the experiment, to allow the current signals obtained to be converted into DA concentrations. All signals used in the analysis matched the expected voltage-current curve for DA [48].

### 4.5. Statistical Analysis

All data are shown as mean ± SEM values and were analyzed using an unpaired t-test or a one- or two-way analysis of variance (ANOVA) followed by a Bonferroni post hoc test for multiple comparisons using appropriate software (Prism 6.02; GraphPad, San Diego, CA, USA). A *p*-value <0.05 was considered statistically significant.

## 5. Conclusions

There is a large body of data showing that L-DOPA-induced dyskinesia is a function of the degree of striatal DA denervation as well as the potential reorganization of other striatal circuits. This information is summarized in several excellent review articles [49,50]. We hypothesize that by slowing the development of the PD phenotype [22], the development of LID in MP mice is reduced only with early treatments.

## Figures and Tables

**Figure 1 ijms-24-04687-f001:**
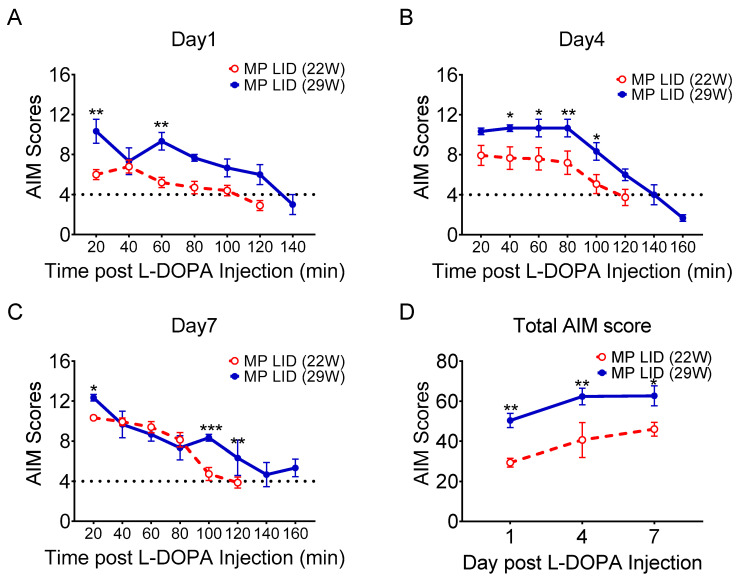
Effect of L-DOPA treatment on MitoPark (MP) of different ages. Comparison of AIM scores between mice that received early PT320 treatment (treatment begun at 5 weeks old and AIM recorded at 22 weeks old) and late treatment (treatment begun at 17 weeks old and AIM recorded at 29 weeks old). AIM scores at (**A**) 1, (**B**) 4, and (**C**) 7 days of L-DOPA/carbidopa treatment in 22-week- and 29-week-old MP mice as well as overall AIM scores (**D**). (**A**) AIM scores in the late treatment group were higher than those in the early group at almost all time points. (**B**) AIM scores were higher in the late treatment group at the middle and late recording time points on day 4. (**C**) Abnormal dyskinesias were longer in the late treatment group. (**D**) Total AIM scores were higher in the late treatment group from day 1 to day 7 after induction protocol. Two-way analysis of variance (ANOVA) followed by Bonferroni post hoc test for multiple comparisons. * *p* < 0.05, ** *p* < 0.01, *** *p* < 0.001 compared with MP LID (22W).

**Figure 2 ijms-24-04687-f002:**
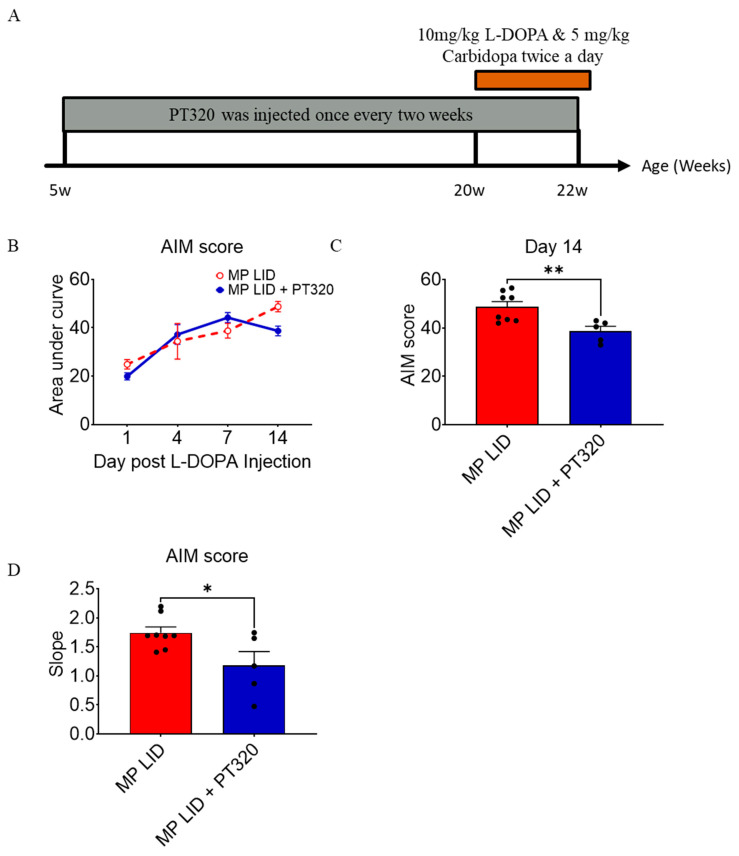
The early administration of PT320 ameliorated the severity of dyskinesia induced by L-DOPA injection after a 2-week induction protocol in 20-week-old MP. (**A**) Early PT320 treatment was started at 5 weeks of age in MP mice while the LID induction protocol was initiated in MP mice at 20 weeks of age. (**B**) The severity of dyskinesia was measured via AIM scores. (**C**) The AIM scores of the PT320 treatment group showed less severe dyskinesia than those of the non-treatment group (MP LID group). (**D**) The slopes of the AIM scores in the PT320 treatment group were less than those of the non-treatment group. Two-way analysis of variance (ANOVA) followed by Bonferroni post hoc test for multiple comparisons * compared with MP LID. * *p* < 0.05, ** *p* < 0.01 compared with MP LID. MP LID (N = 8) MP LID + PT 320 (N = 5).

**Figure 3 ijms-24-04687-f003:**
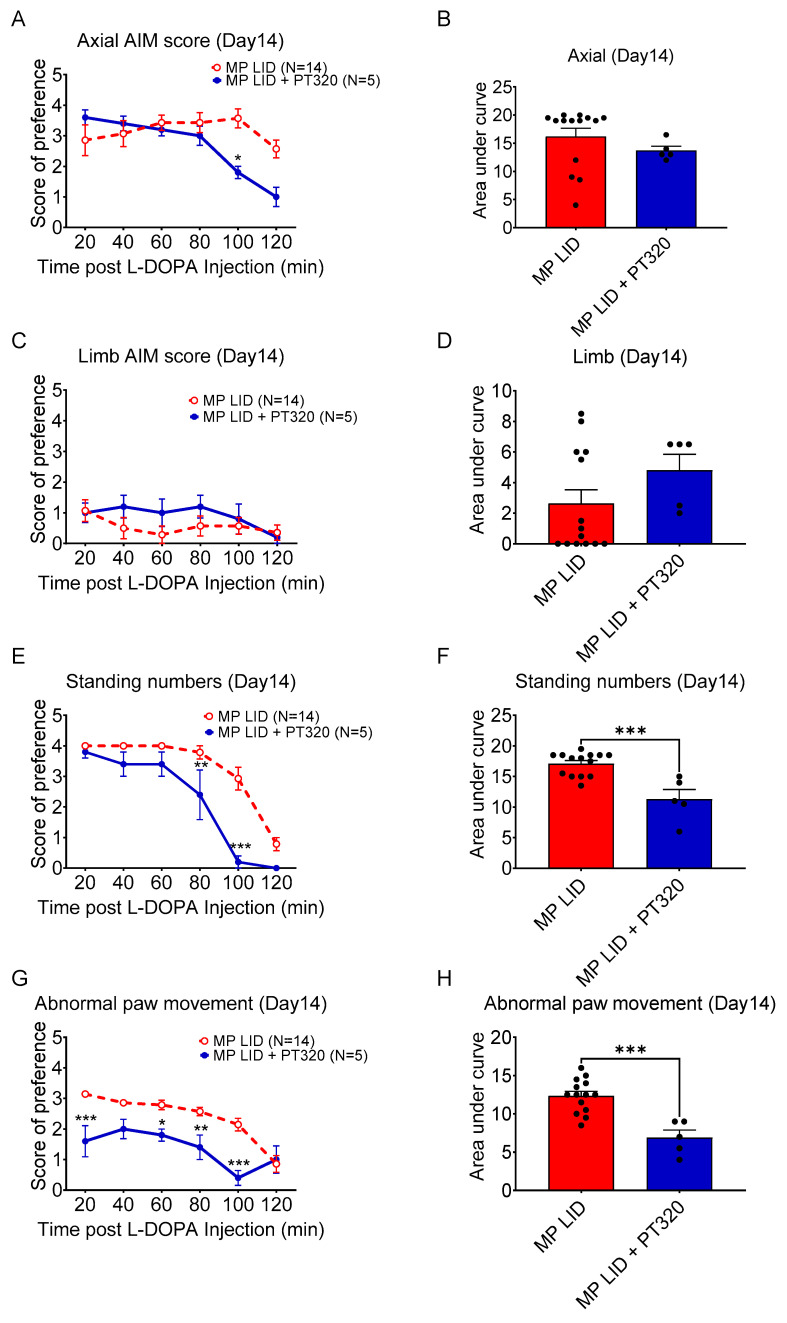
The detailed AIM score components were analyzed between the early 5W PT320 treatment and non-treatment groups at Day 14 of LID induction. (**A**) The axial scores revealed that abnormal axial deviation decreased to normal at the last recording time (**B**) The area under the curve of the scores revealed less severe dyskinesia in the PT320 treatment group compared with the non-treatment group (MP LID group). (**C**,**D**) There were no significant differences in limb AIM scores between groups. (**E**) Abnormal standing time decreased significantly in the PT320 treatment group with less area under the curve in the PT320 treatment group (**F**). (**G**) The score for abnormal paw movement was less in the PT320 treatment group with a significantly lower area under the curve in the PT320 treatment group (**H**). Two-way analysis of variance (ANOVA) followed by Bonferroni post hoc test for multiple comparisons. * *p* < 0.05, ** *p* < 0.01, *** *p* < 0.001 compared with MP LID.

**Figure 4 ijms-24-04687-f004:**
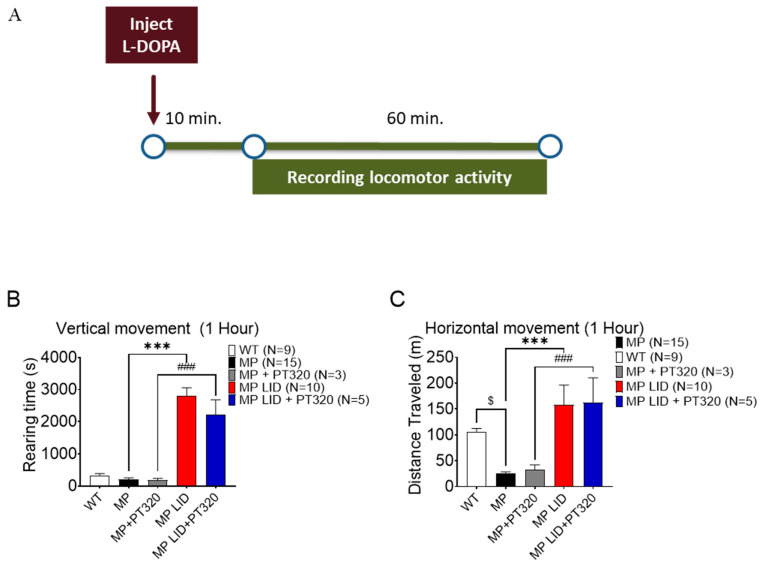
Effect of PT320 combined with L-DOPA on MP locomotion. Locomotor assessments showed no significant differences between groups. (**A**) Experimental protocol for locomotor assessment: one hour of behavioral recording began 10 min after L-DOPA injection. (**B**) There were no significant differences in vertical movement between the MP LID and MP LID with PT320 treatment groups (red bar for MP LID vs. blue bar for MP LID with PT320). (**C**) There were no significant differences in horizontal travel distance between the MP-LID and MP LID with PT320 treatment groups. One-way analysis of variance (ANOVA) followed by Bonferroni post hoc test for multiple comparisons. *** *p* < 0.001 compared with MP LID, ### *p* < 0.001 Compared with MP + PT320, $ *p* < 0.05 compared with WT.

**Figure 5 ijms-24-04687-f005:**
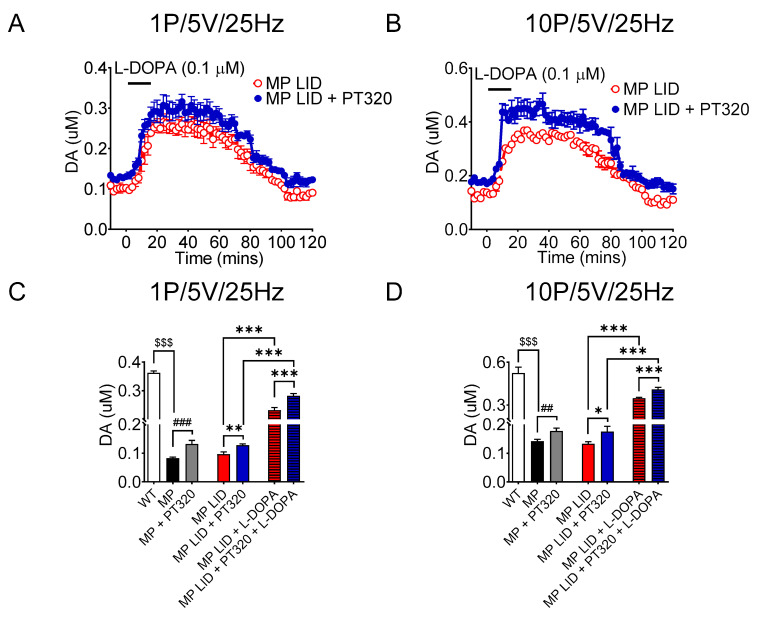
Evoked dopamine release measured via FSCV was augmented in the early PT320 treatment group. (**A**) The tonic release measured via FSCV in the PT320 treatment group in striatal slices after levodopa infusion showed that the PT320 treatment group (blue circle; MP LID + PT320 group) released higher DA concentrations than the non-treatment group (red open circle; MP LID group). (**B**) Bursting (phasic) release change was similar to tonic release after levodopa infusion in the PT320 treatment group. (**C**) The average dopamine tonic release concentration measured via FSCV with or without levodopa infusion was higher in the PT320 treatment group than in the non-treatment group. (**D**) Similar to tonic release, bursting release concentrations were also higher with PT320 treatment. One-way analysis of variance (ANOVA) followed by Bonferroni post hoc test for multiple comparisons. * *p* < 0.05, ** *p* < 0.01, *** *p* < 0.001 compared with MP LID, ## *p* < 0.01, ### *p* < 0.001 compared with MP, $$$ *p* < 0.001 compared with WT.

**Figure 6 ijms-24-04687-f006:**
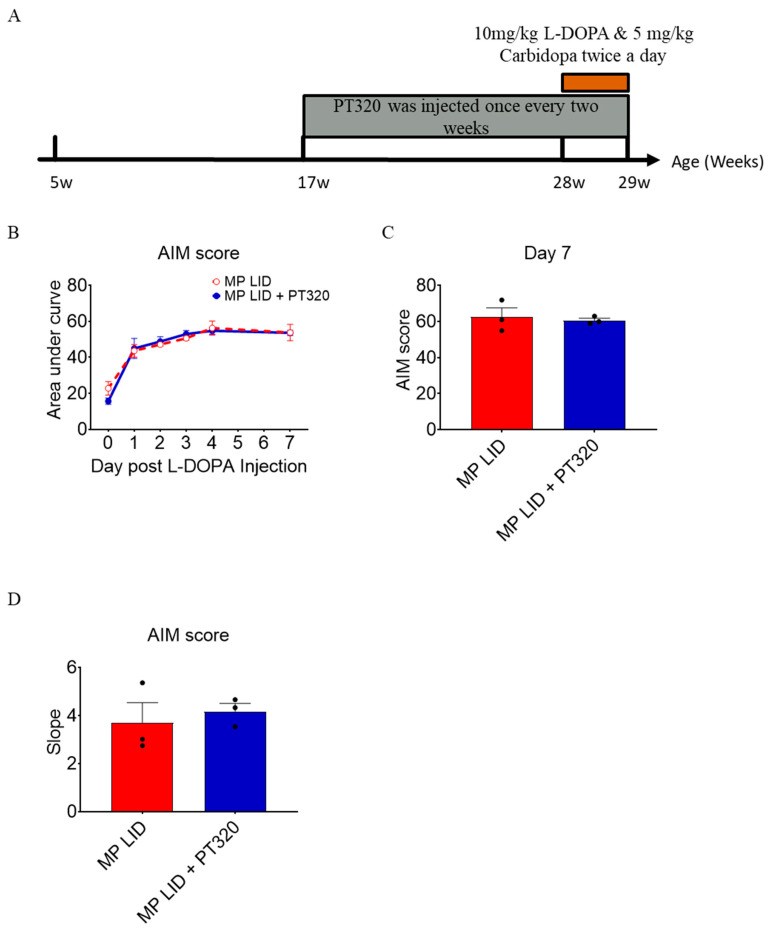
In the late treatment group, PT320 was administered to MP mice beginning at 17 weeks of age and dyskinesia was induced with a one-week induction protocol at 28 weeks of age. (**A**) Experimental design: PT320 treatment in MP started at 17 weeks of age and LID was induced with a one- week protocol at 28 weeks of age. (**B**) There were no significant differences between treatment and non-treatment groups. (**C**) There were no significant differences between AIM scores on day 7. (**D**) The slope of AIM scores also revealed no significant differences between groups.

**Figure 7 ijms-24-04687-f007:**
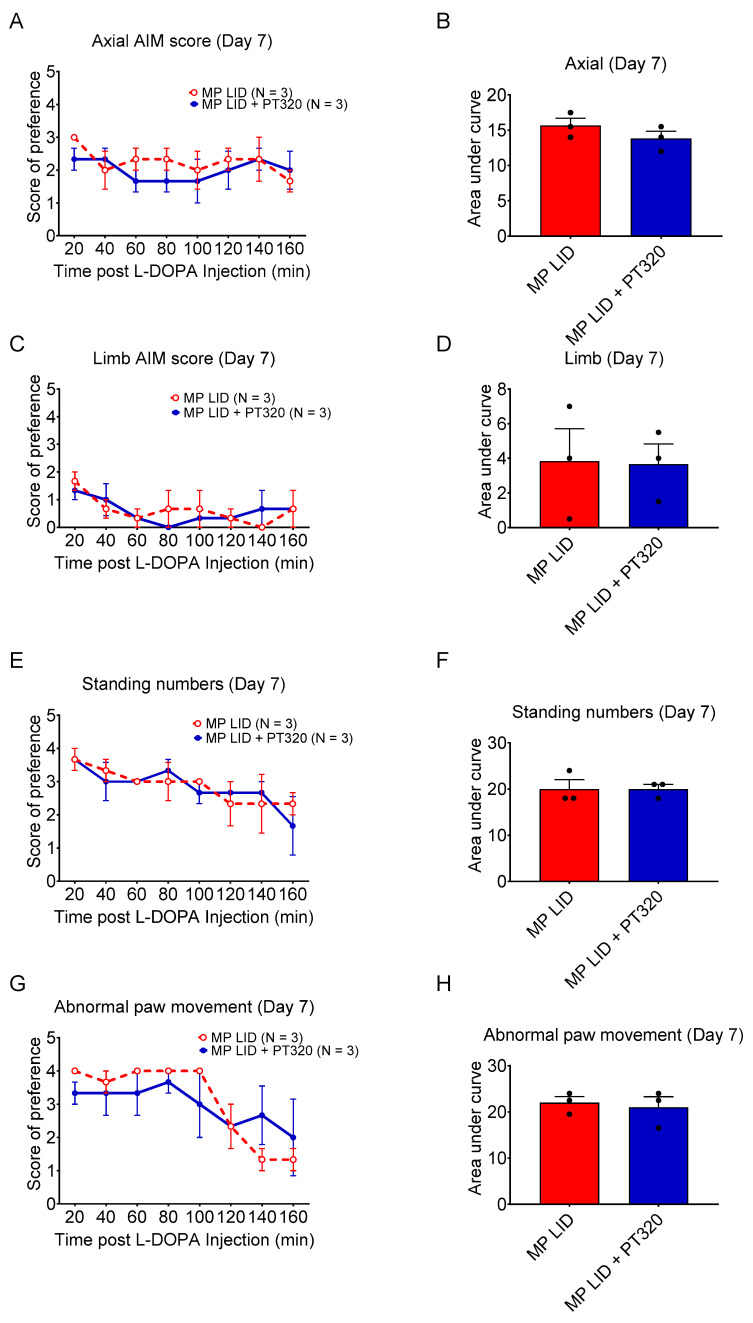
AIM scores on day 7 of induction protocol in the late treatment group. The AIM scores recorded on day 7 of the induction protocol for the late treatment group show no significant difference in (**A**) axial deviation, (**C**) limb abnormal movement, (**E**) standing and (**G**) abnormal paw movement after levodopa injection the between late PT320 treatment group and the non-treatment group. The areas under the curve of each item also showed no significant differences between the groups (**B**,**D**,**F**,**H**).

**Figure 8 ijms-24-04687-f008:**
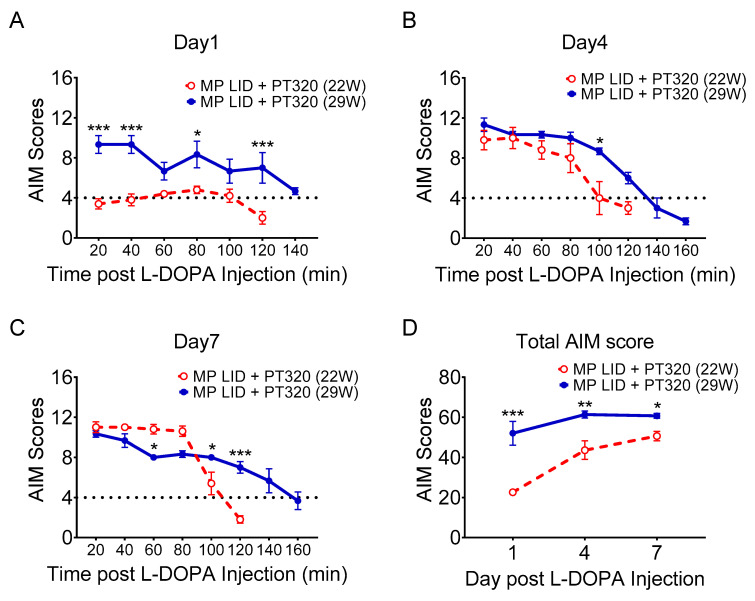
AIM scores comparing the early PT320 treatment (treatment begun at 5 weeks old and AIM recorded at 22 weeks old) and late PT320 treatment groups (treatment begun at 17 weeks old and AIM recorded at 29 weeks old). (**A**) AIM scores in the late treatment group were higher than in the early treatment group at almost all recording time points. (**B**) AIM scores were higher in the late treatment group at the middle and late recording time points on Day 4. (**C**) The duration of abnormal dyskinesia was longer in the late treatment group. (**D**) Total AIM scores were higher in the late treatment group from day 1 to day 7 of the induction protocol than scores in the early treatment group during a similar period. Two-way analysis of variance (ANOVA) followed by Bonferroni post hoc test for multiple comparisons. * *p* < 0.05, ** *p* < 0.01, *** *p* < 0.001 compared with MP LID + PT320 (22W).

## Data Availability

No new data were created or analyzed in this study. Data sharing is not applicable to this article.

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
