# Peer review of "PT320, a Sustained-Release GLP-1 Receptor Agonist, Ameliorates L-DOPA-Induced Dyskinesia in a Mouse Model of Parkinson’s Disease"

_ijms, 2023, doi:10.3390/ijms24054687_

Round 1

Reviewer 1 Report (Previous Reviewer 1)

I accept the manuscript

Author Response

Thanks for reviewer 1 recommands.

Reviewer 2 Report (New Reviewer)

The manuscript covers experimental data on effects of GLP-1 receptor agonist on L-DOPA-induced dyskinesia (LID) in mice with the Parhkinson's phenotype. Development of LID is a well-known problem in long-term treatments of Parkinson disease, therefore any attempt to minimize the adverse effects of L-DOPA is important for improving the outcome of the  treatment. In general, the manuscript is written logically. It correctly presents all the data obtained. The quality of figures is rather high. However, in the section Discussion, the authors should also focus on various mechanisms of  the anti-LID action of the tested compund (in addition to its possible protective action on dopaminergic neurons, GLP-1 receptors agonists can support neurogenesis, control the permeability of blood-brain barrier to prevent progeression of neuroinflammation, etc.). Recent data suggest that LID develops as a multisystem condition affecting many brain areas, so, are there any other targets for the anti-LID activity of PT320? 

Author Response

Reviewer 2 :However, in the section Discussion, the authors should also focus on various mechanisms of  the anti-LID action of the tested compund (in addition to its possible protective action on dopaminergic neurons, GLP-1 receptors agonists can support neurogenesis, control the permeability of blood-brain barrier to prevent progeression of neuroinflammation, etc.).

A: Thanks for reviewer's recommends. By far, the mechanism of GLP-1 receptor agonist is still elusive. In past our study, we could demonstrate GLP-1 receptor potentiate cell viability via  the protein kinase cAMP-dependent (PKA) and phosphoinositide 3-kinase (PI3K)-AKT pathways. Therefore, we argue that anti-LID effect of GLP-1 receptor agonist attributes from its protecting dopaminergic  neuron from apoptosis and keeping normal release of dopamine. 

 Recent data suggest that LID develops as a multisystem condition affecting many brain areas, so, are there any other targets for the anti-LID activity of PT320? 

A:  In the discussion (line 340-344), we have addressed that beneficial GLP-1R mediated actions translate to humans with actions being mediated via brain insulin-signaling proteins, Akt and mechanistic target of rapamycin (mTOR) pathways.  

Reviewer 3 Report (New Reviewer)

1. Justify why MitoPark mouse model is selected for THIS type of study. I believe it is not appropriate model to check the interest of the paper

2. Other results were well described including methodology

3.  This interesting study requires  mRNA, Protein expression, and biochemical assay analysis to concrete the results

4. Check the authenticity of Figure 8, panel D results

Author Response

Reviewer 3:

 Justify why MitoPark mouse model is selected for THIS type of study. I believe it is not appropriate model to check the interest of the paper

A: Thanks for reviewer’s command! In this work, we want to clarify that MitiPark mice, a genetic model of PD, exhibit dyskinesia side effect after 2-week L-DOPA treatment. I DO NOT think any animal model can completely represent human PD, to say nothing of neurotoxin-induced hemiparkinsonian rats!

3.  This interesting study requires  mRNA, Protein expression, and biochemical assay analysis to concrete the results

A:  Thanks for reviewer’s command! In this work, we adopted electrochemistry (Echem) to analyze the quantum release of dopamine in MitoPark mice with/without treatment of PT320. In the future, we will use RNA-seq, LC/MAS to further investigate the protein or gene expression change.

4. Check the authenticity of Figure 8, panel D results.

A:  Thanks for reviewer’s command! We sincerely make sure that Figure 8 panel D results is correct!

Round 2

Reviewer 3 Report (New Reviewer)

all the comments were addressed satisfactorily

This manuscript is a resubmission of an earlier submission. The following is a list of the peer review reports and author responses from that submission.

Round 1

Reviewer 1 Report

The manuscript "PT320, Sustained-Release GLP-1 Receptor Agonist, ameliorates 2 L-DOPA-Induced Dyskinesia in a Mouse Model of Parkinson's 3 Disease" is a search for more effective treatment of the side effects of L-dopa.

Strengths:

The choice of the subject of the study of dyskinesias in PD is strongly justified. PD patients with dyskinesia often do not respond to medication and have a very reduced quality of life.

Weaknesses:

Advanced phase PD strategy is not included.

My comments:

Introduction:

….L-DOPA (L-3,4-dihydroksyfenyloalanina) jest uważana za standardowe leczenie choroby Parkinsona (PD); jednakże przewlekÅ‚e podawanie czÄ™sto powoduje nieprawidÅ‚owe ruchy mimowolne (AIM), zwane także dyskinezami wywoÅ‚anymi przez L-DOPA (LID), u pacjentów z PD…..

This sentence should be completed. The treatment strategy for PD is to administer L-dopa, which is effective for the first few years. In some patients, it leads to di dyskinesia, fluctuations after several years of therapy.

Materials:

Young adult mice are 8-12 weeks old.

Animals 5-98 weeks old were used in the experiment.

5-month-old animals may have developmental changes, not pathology, while 98-week-old senile changes, e.g. as in people over 65 years of age.

Methods

There is no group of young adult animals that were given nothing as a control.

Results

It is a pity that the authors did not confirm neuropathological changes in the CNS on sections or in the TUNEL method.

Discussion:

The authors could propose a probable mechanism of early and late changes in the context of the clinic and provide biochemical parameters that could become targets for the action of drugs in the future. It would be worth mentioning the treatment of advanced phase PD, which is not based on L-dopa but DBS, duodopa, and apomorphine.

Correct editorial errors.

The manuscript "PT320, Sustained-Release GLP-1 Receptor Agonist, ameliorates 2 L-DOPA-Induced Dyskinesia in a Mouse Model of Parkinson's 3 Disease" is a search for more effective treatment of the side effects of L-dopa.

Strengths:

The choice of the subject of the study of dyskinesias in PD is strongly justified. PD patients with dyskinesia often do not respond to medication and have a very reduced quality of life.

Weaknesses:

Advanced phase PD strategy is not included.

My comments:

Introduction:

….L-DOPA (L-3,4-dihydroksyfenyloalanina) jest uważana za standardowe leczenie choroby Parkinsona (PD); jednakże przewlekÅ‚e podawanie czÄ™sto powoduje nieprawidÅ‚owe ruchy mimowolne (AIM), zwane także dyskinezami wywoÅ‚anymi przez L-DOPA (LID), u pacjentów z PD…..

This sentence should be completed. The treatment strategy for PD is to administer L-dopa, which is effective for the first few years. In some patients, it leads to di dyskinesia, fluctuations after several years of therapy.

Materials:

Young adult mice are 8-12 weeks old.

Animals 5-98 weeks old were used in the experiment.

5-month-old animals may have developmental changes, not pathology, while 98-week-old senile changes, e.g. as in people over 65 years of age.

Methods

There is no group of young adult animals that were given nothing as a control.

Results

It is a pity that the authors did not confirm neuropathological changes in the CNS on sections or in the TUNEL method.

Discussion:

The authors could propose a probable mechanism of early and late changes in the context of the clinic and provide biochemical parameters that could become targets for the action of drugs in the future. It would be worth mentioning the treatment of advanced phase PD, which is not based on L-dopa but DBS, duodopa, and apomorphine.

Correct editorial errors.

Author Response

Reviewer 1, general comments:

Advanced phase PD.

We thank Reviewer 1 for checking the “yes” box for all the critique categories. The overall Reviewer’s comment is now clarified in the Introduction as 28-29 weeks old MP mice, which are equivalent to stage 4-5 PD in humans. 

Specific Points:

Introduction

Incomplete sentence.

  1. We are not sure which sentence needs to be completed since this point is not in English. However, we have revised the very first sentence starting with L-DOPA to make it more correct grammatically.

Method

Ages of MP mice used in this study.

  1. We agree that young adult mice are 8 weeks old, however, as we have previously reported 1, 2 the neuropathological changes in MP mice are first seen at 5 weeks and that is our rationale for starting PT320 at that age in the early treatment group rather than 8 weeks.

No group of MP mice.

  1. We’re not sure where the comment about 98 weeks old mice came from. The late PT320 treatment group was started at 17 weeks and animals were studied at 28-29 weeks, not 98 weeks. Mice older than 30-31 weeks show progressive debility which confounds their study and weight loss.

Methods and Results

Neurophysiological changes.

  1. We have extensively studied the progression of the PD phenotype in MP mice in our previous papers which are cited in the reference section of this paper 1, 2. These studies involved behavior, DA electrophysiology, DA electrochemistry, histochemistry, and biochemistry. Since MP mice are subfertile, we can only generate them by matings between heterozygotes and genotyping the offspring, which are one forth of the litter at best (analogous to Mendel’s pea plants). We have thus not included young untreated MP mice as a control and have now explained this in the Discussion [Page 21, line 303].

Discussion

Mechanisms of early and late changes and novel therapeutic targets.

  1. This section is revised to strengthen the use of incretin (GLP-1 and GIP) as future drug therapy targets based on the data in this paper as well as our previous studies 3, 4, on PT320 as the Reviewer emphasized.

Reviewer 2 Report

The purpose of this study was to investigate the effects of PT320 on L-DOPA induced dyskinesia behavior and neurochemistry in a progressive Parkinson’s disease (PD) MitoPark mouse model. However, there are some problems.

1.       In the introduction, the author should briefly introduce how L-DOPA administration can induce LID in MitoPark mice or cite references. If there is no reference, the author should design experiments to confirm that the L-DOPA administration method used in this article can induce LID, which is the basis of this article.

2.       In the aged mice, the AIM score may increase with the increase of mouse month age. Therefore, in Figure 1, MitoPark mice aged 22 and 29 weeks without L-DOPA and the littermate wild type mice should be used as controls.

3.       LID is not suitable for using the word "expression". In line 196, "20-week-old MitoPark littermates", 20 week old mice and 28 week old mice are not littermates.

4.       In Figure 2, Figure 3, Figure 6 and Figure 7, the conclusion that early treatment with PT320 can improve LID in MitoPark mice cannot be reached only through the experiments of MP LID and MP LID+PT3202 groups. It has been reported that PT320 can improve PD progression in MitoPark mice.

5.       There is a grammatical error in the sentence on line 387.

Author Response

Reviewer 2

Mechanisms of how L DOPA induces LID.

  1. We are not sure why Reviewer 2 raises the issue of substantiating why L-DOPA treatment induces LID. There are a large number of studies to show this. We have added two excellent review articles as new references 5, 6 in the Conclusion section to further emphasize this causative relationship between L DOPA and LID.

Untreated MP mice, see comment 2, Reviewer 1.

  1. As noted in our response to Reviewer 1, MP mice are sub fertile and heterozygotes must be mated to produce genotyped MP mice with one fourth of the litter at best. We have not seen MP mice at 22 and 29 weeks with LID changes without L DOPA treatment 1, 3. Thus, given the limited number of MP mice generated in our breeding facility, we have chosen not to use untreated MP mice and WT littermate controls. We have used these groups as controls in our previous paper on PT320 treatment of MP mice 3.

“Expression” and “littermates” in first paragraph of results.

  1. The work “expression” has been deleted in Results line 5. The 20 and 28 week old MP mice are indeed littermates since MP mice from the same litter were often treated with PT320 starting at 5 weeks and 17 weeks. However, we have deleted the term littermates in this section.

See responses above about no untreated MP mice and WT littermates.

  1. Indeed this is the major point of our study. Since PT320 slows the progress of the PD phenotype, as shown by our behavioral electrochemical, histochemical and biochemical measures, and since a major mechanism for L DOPA-induced LID is the level of DA denervation, the slowing of PD phenotype in our primary hypothesis about why PT320 reduces L DOPA induced LID if given prior to the significant loss of the nigrostriatal DA pathway. As noted above, the limited number of MP mice that are generated precluded other experimental groups, which we have previously shown 1, 3 (Ref) do not manifest dyskinesia without L DOPA administration even in late ages.

Comment about grammatical error.

  1. The grammatical error in former line 301 has been corrected to be “in striatal slices from MitoPark mice 3 but also in L DOPA treated animals”.

Round 2

Reviewer 2 Report

1.      In the introduction, the author should briefly introduce how L-DOPA administration can induce LID in MitoPark mice or cite references. If there is no reference, the author should design experiments to confirm that the L-DOPA administration method used in this article can induce LID, which is the basis of this article.

The authors did not answer the question correctly. The authors need to confirm that L-DOPA administration can induce LID in MitoPark mice.

2.      In the aged mice, the AIM score may increase with the increase of mouse month age. Therefore, in Figure 1, MitoPark mice aged 22 and 29 weeks without L-DOPA and the littermate wild type mice should be used as controls.

Correct controls are still required, otherwise the conclusions cannot be drawn, which is the basic principle of designing experiments.

3.      LID is not suitable for using the word "expression". In line 196, "20-week-old MitoPark littermates", 20 week old mice and 28 week old mice are not littermates.

Some places still haven't been corrected.

4.      In Figure 2, Figure 3, Figure 6 and Figure 7, the conclusion that early treatment with PT320 can improve LID in MitoPark mice cannot be reached only through the experiments of MP LID and MP LID+PT3202 groups. It has been reported that PT320 can improve PD progression in MitoPark mice.

The question was not answered or resolved.

Author Response

Specific responses to Reviewer 2: (These are in bold type after each of Reviewer 2’s comments)

  1. In the introduction, the author should briefly introduce how L-DOPA administration can induce LID in MitoPark mice or cite references. If there is no reference, the author should design experiments to confirm that the L-DOPA administration method used in this article can induce LID, which is the basis of this article.

The authors did not answer the question correctly. The authors need to confirm that L-DOPA administration can induce LID in MitoPark mice.

The L-DOPA protocol used here in MP mice is similar to our earlier paper to elicit LID (Refs 12, 13, 14), which was addressed in the Introduction (lines 83-87). Also, we added a sentence about this in the “Methods” section [lines 404-406]. In addition, there is a huge literature on L-DOPA-induced LID mechanisms and we have cited two excellent reviews (Refs 53, 54) on this. Finally, to restate the obvious, this regimen of L-DOPA elicited LID in the present study, which was reduced in the early PT320 administration group.

  1. In the aged mice, the AIM score may increase with the increase of mouse month age. Therefore, in Figure 1, MitoPark mice aged 22 and 29 weeks without L-DOPA and the littermate wild type mice should be used as controls.

Correct controls are still required, otherwise the conclusions cannot be drawn, which is the basic principle of designing experiments.

We thank the reviewer for this comment.  If Reviewer 2 has a reference to aged MP mice showing AIM without L-DOPA, we would appreciate the citation. Therefore, we included the WT, MP and MP+PT320 groups to determine the AIM scores at 22-week-old and 29-week-old.  As shown in Line 122-129 and supplement Figure 2, MP mice at different ages without L-DOPA treatments, have no evidence of increased AIM. On the contrary as noted above and in our reference citations, the MP mice exhibit progressively severe bradykinesia and debilitation followed by L-DOPA treatment necessitating sacrifice for ethical reasons with advanced age. The purpose of data shown in Fig 1 was to demonstrate L-Dopa effects on AIM scores days 1, 4 and 7, on 22- and 29-week-old MP mice. The figure shows a robust significant difference between 22 and 29 week old mice, demonstrating the age-related role for AIM scores. We kindly state that the purpose of the figure has been well demonstrated.

  1. LID is not suitable for using the word "expression". In line 196, "20-week-old MitoPark littermates", 20 week old mice and 28 week old mice are not littermates.

Some places still haven't been corrected.

We thank the reviewer for this comment. We have changed "LID expression" to "LID".

We are perplexed about Reviewer 2’s objection to the term “littermates”. As detailed in the “Methods”, breeding MP mice involves heterozygote crosses which yield about one fourth of each litter as MP mice (Line 373-386). In this paper, we treated some of the MP mice from a given litter at 5 w with PT320 and at 20 w with L-DOPA. Other MP mice from the same litter were treated at 17 w with PT320 and 28 w with L-DOPA. Thus, some of the early and late treatment group animals are indeed littermates (Line 389-397). However, we did delete the terms “littermates” and “expression” from this paper given Reviewer 2’s objection to these terms.

  1. In Figure 2, Figure 3, Figure 6 and Figure 7, the conclusion that early treatment with PT320 can improve LID in MitoPark mice cannot be reached only through the experiments of MP LID and MP LID+PT3202 groups. It has been reported that PT320 can improve PD progression in MitoPark mice.

The question was not answered or resolved.

It is not clear what Reviewer 2’s objections are. Indeed, our hypothesis is that by slowing the development of the PD phenotype by early administration of PT320, we reduce development of LID. This is clearly stated in the Conclusion paragraph. We respectfully state that the strong statistically significant effects in figure 5 and the lack of statistical effects in Figures 6 and 7 together strongly supports our statements.
